# Gene Editing Technologies to Target HBV cccDNA

**DOI:** 10.3390/v14122654

**Published:** 2022-11-28

**Authors:** Maria Guadalupe Martinez, Elena Smekalova, Emmanuel Combe, Francine Gregoire, Fabien Zoulim, Barbara Testoni

**Affiliations:** 1INSERM U1052, CNRS UMR-5286, Cancer Research Center of Lyon (CRCL), 69008 Lyon, France; 2Beam Therapeutics, Cambridge, MA 02142, USA; 3Hospices Civils de Lyon (HCL), 69002 Lyon, France; 4Université Claude-Bernard Lyon 1 (UCBL1), 69008 Lyon, France

**Keywords:** hepatitis B virus, HBV, cccDNA, gene editing, meganuclease, TALEN, ZNF, CRISPR/Cas, base editing

## Abstract

Hepatitis B virus (HBV) remains a significant cause of mortality and morbidity worldwide, since chronic HBV infection is associated with elevated risk of cirrhosis and hepatocellular carcinoma. Current licensed therapies against HBV efficiently suppress viral replication; however, they do not have significant effects on the intrahepatic covalently closed circular DNA (cccDNA) of the viral minichromosome responsible for viral persistence. Thus, life-long treatment is required to avoid viral rebound. There is a significant need for novel therapies that can reduce, silence or eradicate cccDNA, thus preventing HBV reemergence after treatment withdrawal. In this review, we discuss the latest developments and applications of gene editing and related approaches for directly targeting HBV DNA and, more specifically, cccDNA in infected hepatocytes.

## 1. Introduction

### 1.1. Introduction to HBV Biology

Hepatitis B virus (HBV) remains a significant cause of mortality and morbidity worldwide [1]. Chronic HBV infection (CHB) has a poor prognosis due to cirrhosis and hepatocellular carcinoma (HCC) [2]. Current licensed therapies against HBV, including nucleoside analogs (NAs) and pegylated interferon (Peg-IFN), suppress viral replication. However, neither of these therapies have significant effects on the covalently closed circular DNA (cccDNA) of the viral minichromosome responsible for viral persistence, and thus life-long treatment is required [3,4].

cccDNA is formed from the relaxed circular DNA (rcDNA) delivered by the viral particle [5]. Once established, cccDNA is the only template for viral pregenomic RNA (pgRNA) and all the viral subgenomic mRNAs. There are four overlapping open reading frames (ORFs) in cccDNA: surface (S), precore/core (C), polymerase (P) and X ORFs. These four ORFs encode seven viral proteins: HBeAg (secreted protein), HBc (viral capsid protein), HBV POL/RT (polymerase reverse transcriptase), PreS1/PreS2/HBsAg (large, medium and small surface envelope glycoproteins) and HBx (transcriptional activator). pgRNA is retrotranscribed by the viral polymerase into rcDNA. A fraction of rcDNA is packaged and released, forming new infectious particles (i.e., Dane particles), while some will be transported to the nuclei to replenish the cccDNA pool. Thus, cccDNA serves as the molecular reservoir that is key to HBV persistence: only a few copies of cccDNA are sufficient for infection rebound after treatment cessation [6]. Additionally, HBsAg and a truncated but potentially functional HBx form can be produced from integrated HBV DNA [7,8] (Figure 1). HBV DNA integration causes alterations in the host genome and has been suggested to be an important factor contributing to the development of HCC [9]. However, it is still debatable whether the removal of integrated HBV DNA from the host genome is a necessary measure to recover chromosomal stability and cure HBV-related HCC.

Therapies aiming to achieve functional HBV cure, defined as the undetectability of HBsAg and DNA in serum, with or without HBs seroconversion, are actively being evaluated [10]. However, a functional cure does not aim for cccDNA elimination from the infected hepatocytes, though it may require its silencing and/or inactivation. Instead, a complete HBV cure, aiming for cccDNA elimination, or a sterilizing cure, leading to both cccDNA and integrated HBV DNA elimination, are desired. Undeniably, therapies that can lead to the reduction, silencing or eradication of cccDNA to prevent HBV reemergence after treatment withdrawal are sorely needed [4,6]. In this section, we discuss the latest developments and applications of gene editing and related approaches to targeting HBV DNA and, more specifically, cccDNA.

### 1.2. Introduction to Gene, Base, Prime Editing, CRISPRa and CRISPRi

Advances in the gene editing field have given rise to a wide range of tools available for various applications. Here, we will review the technologies that are relevant and/or which have been used for HBV genome targeting.

Meganucleases (also called homing endonucleases or rare-cutting endonucleases) are endodeoxyribonucleases with large recognition sites (14–40 nt), providing high specificity for genome editing applications [11]. A meganuclease system generates a double-stranded break (DSB) that can be processed by cell repair machinery either through the non-homologous end-joining path (NHEJ), leading to indels and knockout of the gene in question, or through homologous directed repair (HDR), which requires the presence of the DNA template and allows the correction of genomic DNA. Given their conveniently small size (1 kb), meganucleases are currently being explored for a number of applications [11,12]. Their downside is that they are difficult to reprogram to recognize different sites; thus, they were later replaced by zinc-finger nucleases (ZFNs) and transcription-activator-like effector nucleases (TALENs). Both have a DNA cleavage domain fused with a DNA-binding domain, which has a modular structure and can be reprogrammed faster to recognize the site of interest [13,14]. ZFNs are composed of the sequence-independent cleavage domain of the restriction enzyme FokI fused to the tandem zinc-finger protein (ZFP) domains. TALENs couple a non-specific FokI nuclease domain to a TALE binding domain (transcription-activator-like effectors), enabling direct and sequence-specific DNA cleavage, with less cytotoxicity than ZFNs.

Currently, the most widely used gene editing system is CRISPR/Cas, which was first described for mammalian gene editing in 2012 [15]. Cas9 is an RNA-guided microbial nuclease that originated from a clustered regularly interspaced short palindromic repeat (CRISPR) bacterial immune system. The Cas9 enzyme is targeted by the guide RNA (gRNA), which contains a short 20 nt sequence complementary to the genomic site of interest which defines the recognition site and the point of DNA cleavage. The genomic DNA site further requires the presence of a protospacer-adjacent motif (PAM)–typically a 2–6 nt sequence. Aside from the requirement of a PAM sequence, Cas9 is easily targetable to new sites simply by changing the gRNA sequence. As with other nucleases, CRISPR/Cas-system-induced DSBs can be processed by NHEJ to generate indels or by HDR to insert a sequence defined by an exogenous DNA template [13,16,17,18] (Table 1).

Base editors (BEs) are alternative gene editing tools that introduce precise mutations in targeted sequences through chemical reactions of deamination without inducing DSBs, thus not requiring donor DNA templates for gene correction. BEs were engineered by fusing partially inactivated Cas nucleases to deaminases. Currently, there are two major classes of base editors: C>T base editors (CBEs), which convert cytosine to thymine [19], and A>G base editors (ABEs), which convert adenine to guanine [20]. Almost all current base editors possess a nickase activity; a nick in the strand opposite to the edited one biases cellular DNA-repair systems to resolve the mismatch into the desired edit. Furthermore, most CBEs contain a uracil glycosylase inhibitor peptide (UGI) that prevents the activation of the base excision repair system. As base editors do not generate DSBs and therefore do not stimulate NHEJ, they generate fewer off-target indel mutations and are more efficient in non-dividing cells compared to the standard nucleases [17,18,19,20]. Glycosylase base editors have been developed more recently through the replacement of UGI with a uracil-DNA glycosylase (UNG) in certain CBEs, which has enabled C>A transversion changes in Escherichia coli and C>G transversion changes in mammalian cells [21]. More studies will be necessary to evaluate their clinical relevance. Another set of recent gene editing tools that can introduce conversion mutations, as well as precise deletions or insertions, are prime editors (PEs). A PE is constituted by a partially inactivated Cas nuclease fused with a reverse transcriptase (RT) domain [22]. PEs directly rewrite desired genomic sites through reverse transcription initiated by Cas9 nickase activity. Prime editing gRNA (pegRNA) not only guides the complex to the site of interest but further encodes the template for the RT reaction. An additional gRNA is necessary to nick the non-edited strand, stimulating repair systems to replicate the modified strand and complete the editing of the genomic site [17,22].

The CRISPR/Cas system has been further optimized for epigenetic regulation of gene expression. CRISPR-based epigenomic regulators consist of an inactivated Cas9 fused to an effector, typically a repressor or an activator of gene transcription [23,24]. Accordingly, the technology is referred to as CRISPR activation (CRISPRa) or CRISPR interference (CRISPRi). The minimal CRISPRi system is assembled by a dead Cas9 with a gRNA, which physically blocks gene expression [25]. Fusing dCas9 further to the repressor domains, such as the Krüppel-associated box (KRAB), significantly improves silencing in mammalian systems through heterochromatin formation at the target site [25,26]. The CRISPRa system can employ various protein effectors, such as the NF-kB p65 subunit, herpes simplex viral protein 16 (VP16) and rationally engineered tripartite transcription activator VP64-p65-Rta (VPR) [27]. An alternative approach to tether a transcription effector to the Cas complex is to include effector-binding sites in the extended gRNA scaffold [28].

## 2. Obstacles and Limitations Regarding the Use of Nuclease-Based Approaches for Antiviral Therapy

### 2.1. Targeted and Efficient Delivery

Compared to small NUCs, gene editing components are large (their sizes range from 1 kb for meganucleases up to 7 kb for prime editors, plus approximately 100 nt of gRNA in the case of RNA-guided nucleases); therefore, their delivery represents a challenge.

Several delivery strategies have been explored for gene editing systems [29]: (1) viral delivery, using, for example, adenoviruses and adeno-associated viruses (AAVs); (2) non-viral methods, utilizing lipid-based nanoformulations (LNPs) of mRNA and gRNA or ribonucleoproteins; (3) physical methods, i.e., electroporation for ex vivo applications, such as editing in hematopoietic stem cells. The choice of method is dependent on the cell/organ type and the application. The two main approaches for the hepatic delivery of gene editing reagents are viral methods and LNPs, which will be further discussed below (Table 2).

#### 2.1.1. Viral Delivery

AAVs are single-stranded DNA viruses that stay in cells as episomal DNA after transduction [30]. AAVs generally do not elicit strong immune responses and have been used in clinical trials for the delivery of gene editing systems since 2016 [31]. In the first clinical trials, AAV-CRISPR technology was applied ex vivo to generate PD-1 knockout T-cells for the treatment of advanced lung cancer [30]. Recently, the first in vivo clinical trial was initiated to assess the safety and efficacy of the AAV5-SaCas9 system used to correct the CEP290 gene for the treatment of a genetic blindness disorder [32,33].

A key limitation of AAVs as viral vectors is their packaging capacity, which is approximately 4.7 kb. While the smaller nucleases (meganucleases, SaCas9 or compact base editors) fit into a single AAV vector, more standard gene editing systems, such as SpCas9, or larger base editors have to be split into two different AAV vectors, for example, via the intein-mediated approach [34,35,36].

The tropism of the AAV serotype used defines the cell/organ that can be targeted. AAV2 and AAV8 serotypes have both been used for hepatic delivery in mice [12,30], in particular, for the inhibition of HBV viral replication in vivo in an HBV-infected humanized mouse model [37].

Concerns about using AAVs include their potential integration into the human genome, resulting in increased likelihoods of mutagenesis and oncogenesis, and the continuous expression of editors [38]. Sustained expression of gene editors can also result in increased rates of off-target editing [38].

Compared with AAVs, adenoviruses (Ads) are able to hold larger DNA insertions of up to 35 kb, so any editing system could fit within the same vector [39]. Previously, a single Ad vector was used to deliver CRISPR/Cas9 and the three gRNAs targeting cccDNA in a HepG2-NTCP-infected cell system [40]. Adenoviruses can target a wide range of cells in an organism, including hepatocytes, with high efficacy. A wild-type Ad5 serotype has a strong affinity for the liver; targeting occurs through the binding of the Ad5 hexon protein to the coagulation factor X [39]. The major downside of using adenoviruses is the immune response that can cause hepatic failure [41,42]. Helper-dependent adenoviruses (HDAds) have large parts of their genome removed and thus do not trigger as strong immune responses. As a result, they maintain long-term expression and are considered to be more clinically relevant.

Lentiviral vectors were used in the past for the expression of gene editing reagents in vitro and in vivo, in particular, for the inhibition of HBV replication in HepG2.2.15 cells via Crispr/Cas9 delivery [43]. While lentiviral vectors have high packaging capacities and support robust transgene expression, mechanistically, they work through DNA integration into the host genome, which is associated with two risks: (1) host genome instability and tumorigenicity and (2) off-target edits due to sustained expression of the gene editing cargo [43]. Integrase-deficient lentiviral vectors are being developed to improve the safety profile of lentiviruses. However, other safer and clinically relevant delivery options exist for hepatocytes, therefore reducing interest in lentiviral use.

#### 2.1.2. Non-Viral Delivery

Non-viral delivery of the gene editing system typically implies using lipid/polymeric materials for the formulation of RNA encoding the gene editing components or, in some cases, protein complexes that directly perform gene editing in nanoparticles. A lipid nanoparticle (LNP) is composed of (1) an ionizable, cationic lipid or polymeric material, which facilitates RNA entrapment in the nanoparticle; (2) a zwitterionic lipid; (3) cholesterol to stabilize the lipid layer; and (4) polyethylene glycol (PEG) to stabilize the LNP in the extracellular space [29,44]. This complex composition enables in vivo targeted delivery through several processes: stabilization of the cargo in the extracellular space; protection against immune system activation; LNP-mediated delivery to a particular cell type; and endosomal escape into the cytoplasm. Ultimately, hepatic delivery occurs through a combination of two mechanisms: passive targeting via endothelial fenestrae that are present in human liver sinusoids [45] and apolipoprotein binding to the LNP, which facilitates liver uptake [29,44].

Compared to viral delivery, LNPs elicit much weaker immune responses in vivo, which translates into their representing a safer technology and also provides potential for redosing the drug several times and increasing the efficacy of the treatment [29,30,44]. Furthermore, the expression of the gene editing component from the mRNA is transient and therefore is associated with lower immunogenicity and fewer off-target effects [44].

LNPs can be optimized for targeting different cell types in vivo; in particular, significant progress has been achieved in targeting hepatocytes [29]. Recent clinical data have shown the safety and efficacy of the gene editing therapy NTLA-2001 designed to treat hereditary amyloidosis through TTR gene knockdown in the liver [46]. NTLA-2001 constitutes a lipid nanoparticle formulated with a Cas9 mRNA and gRNA targeting the TTR gene; a single IV injection at a relatively low dose of 0.1 or 0.3 mg/kg led to a 52–87% reduction in the TTR gene in the blood of the treated patients. These initial data are encouraging for the other gene editing therapies using LNPs for hepatic delivery.

Ribonucleoprotein (RNP) or protein complexes can also be formulated directly into lipid nanoparticles (LNPs) [47]. One of the advantages of this approach, with respect to nuclease mRNA and gRNA LNPs, is the adjusted 1:1 stoichiometry of the editor/gRNA complex [18]; the challenge is to keep the complex active, which can be difficult in terms of manufacturing, as well as the optimization of the protein-LNP formulation for in vivo targeting [44]. RNP LNP delivery has enabled efficient editing in vitro for a number of applications, in particular, for inhibiting HBV replication in infected HepG2-NTCP cells [47]. RNP LNPs further showed high efficacy ex vivo for AsCas12a-mediated editing in human hemopoietic stem cells (Editas). Limited efficacy has been observed in vivo for editing in the lungs and liver [48] and in cochlear cells after direct injection in the inner ears of mice [49,50].

### 2.2. Off-Target Effects

Off-target effects are unintended genetic modifications that occur through the use of gene editing technology. Off-target effects can be gRNA-dependent (i.e., associated with the gRNA sequence recognizing DNA) or -independent (related to the alternative activities of the gene editing enzyme) [18].

In the case of nuclease-mediated editing, a possible gRNA-dependent off-target edit is indel formation, which occurs through the Cas9-induced DSB followed by the NHEJ repair pathway at the potential off-target site. While the 7–12 nt PAM-proximal sequence defines gRNA specificity, the tolerance of the PAM-distal sequence can be quite high. Recognition between the gRNA and the DNA site can happen with 1–3 mismatches and even more in certain cases [51]. gRNA-dependent off-target effects can be reduced by altering gRNA through gRNA modifications, replacing some RNA nucleotides with DNA nucleotides or varying gRNA length [52,53], as well as optimizing the gene editing enzyme [50]. As base editing does not induce DSBs, the number of indels is naturally reduced compared to standard nucleases [19,20]. In the case of a base editor, the expected off-target edit would consist of single nucleotide variations. [18].

Other types of off-target effects are chromosomal deletions and translocations, which could be promoted by nuclease-mediated DSBs [54,55,56]. Unexpected chromosomal truncations can emerge after targeting a genetic locus with Cas9 and can range from several nucleotides up to a megabase scale [37,56]. In actively dividing cells, Cas9-mediated DSBs increased the formation of chromosomal bridges and aberrant nuclear structures, leading to chromothripsis [55]. The single nickase approach can prevent on- and off-target indels in some systems [56].

gRNA-independent off-target effects are associated with the alternative activities of the nuclease. Expression of the Cas9 protein without gRNA can lead to the activation of the p53 pathway and promotes the emergence of p53-inactivating mutations [57]. Mechanistically, it is possible that p53 is activated through increased DNA damage induced by Cas9; studies assessing the abundance and consequences of these effects are currently limited [56,57].

In the case of base editing, the reported unguided off-target activity is mostly associated with the CBE deaminase, which can lead to “spurious deamination” of cellular RNA and, to a very low extent, DNA [58,59]. This effect can be minimized by switching the deaminase or introducing mutations into the enzyme [60]. Recent studies have shown no spurious RNA or DNA deamination in the case of ABE base editors, although more work will be necessary to evaluate ABEs for particular therapeutic applications [18,59,61].

## 3. Application of Gene Editing for the Treatment of Diseases Caused by Episomal Viruses Other Than HBV

Gene editing technologies were successfully applied to target the episomal DNA genomes of several viruses, thus paving the way for similar applications against HBV [62]. Extensive studies have been conducted to develop gene editing approaches for human papillomavirus (HPV) and viruses of the *Herpesviridae* family, such as Herpes simplex virus type-1 (HSV-1) and human Epstein–Barr virus (EBV).

HPVs are relatively small 8 kb dsDNA viruses; they are known to cause almost all cervical carcinomas and some other cancers [63]. Significant progress has been made in applying gene editing for the treatment of HPV-involved cancers through targeting HPV genomes. Nuclease-mediated knockdown of the HPV oncogenic proteins E6 and E7 led to the activation of p53 and retinoblastoma protein genes and cancer cell death in vitro and in vivo [64,65]. A similar approach has been used in clinical trials of CRISPR and TALENs targeting E6/E7 HPV proteins to treat HPV-related cervical intraepithelial neoplasia [30] (NCT03057912).

Herpesviruses are large dsDNA viruses containing 100–200 genes; they can establish lifelong infections in the host and often lead to serious diseases, especially in immune-compromised individuals. HSV-1 infection can lead to viral encephalitis or neonatal herpes in immune-immature or immune-compromised patients [66]. While both meganucleases and CRISPR/Cas systems targeting HPV genomes were efficient in inhibiting HSV-1 infection in various in vitro models, including primary neurons, in vivo delivery in a mouse model proved to be more challenging [66]. A combination of the three different AAV serotypes had to be used to deliver two meganucleases to enable 90% elimination of the HSV-1 genome from the cervical ganglia neurons in a mouse model of HSV ocular infection [67].

EBV genomic episomal DNA is maintained in epithelial and B-cells and can lead to Burkitt’s lymphoma and nasopharyngeal carcinoma [62]. CRISPR-mediated knockdown or depletion of the EBV genome suppressed viral replication in infected lymphoma cell lines in vitro. Different targeting strategies proved to be efficient, including disruption of the EBV nuclear antigen [68], excision of the promoter of the viral microRNAs [69] and a multiplex strategy for the targeting of repeat regions to digest viral genomes into pieces [70].

A limited number of studies are available on the application of nuclease-based approaches for the other types of viruses, such as herpesviruses human cytomegalovirus (HCMV) and Kaposi-sarcoma-associated herpesvirus (KSHV), as well as human polyomavirus 2 (JCV) [62]. While various nuclease-based strategies are efficient in in vitro cell models, in vivo delivery of the editing systems requires further development, especially for viruses infecting cells that are difficult to target, such as neurons.

Altogether, these studies demonstrate that the direct targeting of viral episomal genomes in vitro and in vivo is achievable, with different degrees of efficiency depending on the intrinsic properties of the viral episome, the editing enzyme used and the efficiency of delivery into the organ of interest.

## 4. Permanent Modifications in cccDNA Sequences Affecting HBV Replication

### 4.1. Designer Nuclease and CRISPR/Cas Approaches Targeting HBV DNA

Nucleases could specifically target HBV DNA, inducing double-stranded breaks, which may be repaired by the error-prone non-homologous end-joining repair system (NHEJ), leading to mutations at targeted cleavage sites. Alternatively, episomal cccDNA linearized by DSBs could be susceptible to direct degradation by cellular DNAses, leading to reduced viral replication. An overview of the published studies on HBV editing strategies is given in Table 3.

Though ZFNs showed mild success when used to target cells transfected with an HBV-expressing plasmid [73], no mutations were detected in cccDNA isolated from the cells stably expressing the HBV genome. TALENs reduced viral replication in cell culture and in vivo in a mouse model generated through hydrodynamic injection (HDI) with an HBV-expressing plasmid [71]. When combined with IFN, there was an increased reduction in pgRNA production [72]. TALENs and ZFNs can efficiently disable HBV targets; however, the complexity of their production precludes more efficient screening to select the best product, and direct effect in an established pool of cccDNA in a relevant natural infection model remains to be determined. Recently, data have been reported regarding the use of a specific engineered meganuclease, ARCUS, based on a naturally occurring genome editing enzyme, I-*Cre*I, which evolved in the algae *Chlamydomonas reinhardtii*, to target the HBV genome [92]. Thanks to the small size of the nuclease, clinically relevant delivery was achieved in AAV-HBV mouse and non-human primate models through the systemic administration of LNPs containing ARCUS mRNA, resulting in indel detection, significant reductions in intrahepatic AAV copy numbers in both models and durable serum HBsAg decrease in the mouse model. Moreover, assessment of genomic insertions in host cell genomes was evaluated in primary human hepatocytes, demonstrating an inverse correlation with the specificity of the meganuclease [92].

Lin et al. first reported that dual HBV-specific gRNA/Cas9 cleaved a HBV template [74]. Further, several sites in the HBV genome were validated as targets for a Cas9 antiviral therapy in HDI in vivo models [76,79,86]. Duck hepatitis B (DHBV) in primary duck hepatocytes was also used as a surrogate for HBV infection, showing that dual gRNA targeting reduced HBV total DNA and cccDNA [99]. HBV replication was efficiently reduced by CRISPR-Cas9 in a mouse model carrying an HBV cccDNA-like molecule [75]. Several of these studies suggested HBV DNA degradation when using HBV-expressing plasmids as surrogates of natural infection. However, given the high copy numbers of the plasmids compared to the expected number of cccDNA copies per infected hepatocytes (between 1.4–9.6 copies/cell [100,101]), it is possible that in these conditions the NHEJ system is insufficient or delayed in rejoining the abundant Cas9-induced DSBs in the plasmids, leading to its degradation. Furthermore, the accessibility of these target sites may be different in cccDNA derived from a natural infection. Although CRISPR-Cas9 was able to eventually clear out HBV-expressing plasmids, both in vivo and in cell cultures, these studies should be repeated in infection models comprising authentic cccDNA. CRISPR-Cas9 reduced HBV replication in cell lines with integrated HBV DNA: dual gRNA targeting led to strong reductions in HBsAg/HBeAg secretion and HBV DNA levels, in some cases leading to removal of the sequences between the target sites [74,78,79]. Cell lines carrying integrated HBV DNA do not represent good models for determining cccDNA targeting, since it is challenging to differentiate a direct effect on the episomal cccDNA from an effect on viral integrated sequences that could translate into the indirect appearance of defective cccDNA molecules derived from mutated pgRNA transcribed from edited HBV integrants. Nonetheless, the use of CRISPR-Cas9 in these stably expressing cell lines is of interest regarding the performance of loss-of-function studies: an HBsAg KO cell line was established from HepG2.2.15, PLC/PFR/5 and Hep3B cells, showing impaired HBsAg release and reduced proliferation and growth rates of xenograft HCC tumors [102]. These loss-of-function studies are critical to address the suggested association between high HBsAg levels and HCC [103,104]. Additionally, it was proposed that the removal of integrated HBV DNA from the host genome is necessary to recover chromosomal stability and cure HBV-related HCC [105]. Using HepG2.A64 cells carrying integrated HBV DNA, Li et al. showed that it was possible to eradicate the full-length integrated DNA, preventing viral rebound [85].

Experiments with naturally HBV-infected HepG2-NTCP and HepaRG cells stably expressing gRNA/Cas9 showed that both cccDNA and its precursor rcDNA could be substrates for Cas9, leading to indels in the target sites, rather than cccDNA degradation [93,94]. Interestingly, some gRNA sequences preferentially target integrated HBV DNA, rather than cccDNA, suggesting that some regions of the tightly packed viral minichromosome may not be accessible to gRNA/Cas9 [94]. It is noteworthy that rcDNA and other intermediate species in the rcDNA-cccDNA conversion can carry the gRNA target sequence as dsDNA and can therefore be targeted by gRNA/Cas9. Thus, it is possible that the gRNA/Cas9 complex expressed before infection can target intermediate precursors, leading to truncated cccDNA, rather than directly targeting established cccDNA.

HBV replication was inhibited through mutations introduced with a nickase-Cas9 (Cas9n) combined with a pair of HBV-specific sgRNAs [81,82]. The “paired nicking” strategy introduces single-stranded DNA breaks (nicks) at a specific location; thus, undesired off-target nicks will be precisely repaired using the intact strand as a template [16,106], thereby improving specificity.

In vivo, the transduction of HBV transgenic mice with rAAV8-SaCas9, mainly targeting liver cells, led to variable results. In one study, it led to a reduction in serum HBsAg, HBeAg, HBV DNA and intrahepatic HBcAg [97]. However, a later study did not achieve significant reductions in HBV viral parameters in a hydrodynamically injected (HDI) mouse model, which was attributed to low transduction efficiency in hepatocytes [80]. Though these studies provide proofs of concept that AAV can be used to reduce HBV replication in HBV transgenic mice, these models do not feature complete HBV replication and do not produce cccDNA. Humanized liver mouse models allow HBV replication and cccDNA production, enabling persistent viral infection, and are the in vivo gold standard for the development of HBV antiviral therapies.

Thus, liver-humanized mice were used to evaluate the efficiency of an AAV vector to deliver SaCas9 and HBV-specific gRNAs alone [37] or in combination with entecavir [98]. In this study, five out of eight treated mice had low-frequency mutations in the target sites, which led to HBV infection rebound upon entecavir removal, indicating the need to improve in vivo delivery approaches. Though viral delivery systems have been shown to represent an effective approach to deliver gRNA/Cas9 in hepatocytes in vivo, they are associated with cell toxicity, immunogenicity and long-term Cas9 expression [30]. In this respect, Suzuki et al. developed the production of an LNP-based CRISPR/Cas RNP delivery nanoplatform, which significantly suppressed both HBV DNA and cccDNA in HBV-infected HepG2-NTCP cells compared to AAV2 [47]. Chen et al. explored the capacity of naturally produced exosomes to carry and deliver gRNA/Cas9 functional complexes in HBV-expressing cells. Exosomes carrying two HBV-specific gRNAs and Cas9 protein were evaluated in cells transfected with HBV-expressing plasmids, confirming HBV DNA gene editing [107]. RNP complex delivery leads to transient gene editing, eliminating the risk of DNA integration. However, these results also suggest that the release of these exosomes could affect both surrounding and distal tissues or cells via cell-to-cell transmission, which may further complicate safety profiles, including off-target effects.

Although several studies suggest cccDNA degradation rather than repair after gene editing, it is still controversial whether this would occur in a natural infection in vivo. HBV variants generated after dual gRNA cccDNA targeting were shown to be transcriptionally active in natural HBV infection models, highlighting the importance of understanding the fate of cccDNA after gene editing [95]. Thus, both the HBV infection system and the delivery approach could affect the fate of cccDNA after editing, highlighting the importance of using relevant models of authentic cccDNA to investigate the potential applications of targeted nucleases for HBV [77,87,88]. Additionally, the DNA damage-repair pathways activated in response to CRISPR/Cas treatment would also be involved in determining the fate of HBV genomes. Even if the choice of the repair pathway following CRISPR/Cas DSBs still remains an ill-understood mechanism, it is reasonable to think that NHEJ would be preferentially activated in non-dividing cells, such as hepatocytes. In this respect, the manipulation of the NHEJ system yielded discordant results regarding the capacity for improving cccDNA degradation following CRISPR/Cas9 [108,109], proving the difficulty of modulating or controlling the effects of these complex and redundant DNA damage-repair pathways. The use of different cell culture models with respect to differentiation status and proliferative capacity has, therefore, to be taken into account when comparing results across studies.

### 4.2. Non-DSB Approaches Leading to cccDNA Editing

Standard nucleases act through generating DSBs, which may lead to large deletions and genomic rearrangements, especially in cases where several genomic locations are targeted [54,55,56]. In addition to an episomal cccDNA, chronic hepatitis B patients typically have multiple copies of HBV DNA integrated into their genomes, so the possibility of chromosomal translocations is significantly increased. Newer editing systems, such as base editing and prime editing, modify the genome without generating DSBs, which may present a safer approach for HBV treatment (Figure 2).

The first studies assessed the use of base editing for HBV cure in vitro using C>T base editing (BE3 and BE4 generations) to generate stop codons in the S antigen [91,96]. Naturally HBV-integrated human hepatoma PLC/PRF/5 cells engineered to expresses a CBE (PLC/PRF/5-CBE) were transduced with a lentivirus carrying a sgRNA to introduce a stop codon in the *S* gene. Phenotypically, 71% of cells developed a premature stop codon and HB mRNA levels were significantly decreased, accompanied by a 92% reduction in HBsAg secretion [96]. Lentiviral delivery of the base editing reagents in the HepG2-NTCP system enabled 30–50% cccDNA editing and was associated with reductions in HBsAg and extracellular HBV DNA [96]. These initial data are encouraging; however, assessment of other viral parameters, such as intracellular HBV DNA, RNA and cccDNA levels, will be important to understand the effects of the base editing on the HBV life cycle. Furthermore, using clinically relevant delivery methods, as well as assessing the effect of the base editing on HBV in long-term in vitro systems and mouse models, will be necessary to evaluate the use of the base editing for HBV treatment in patients.

## 5. Transient Modifications Directly Targeting cccDNA or Cellular Factors That Can Target cccDNA

### 5.1. CRISPRa Targeting APOBEC for cccDNA Degradation

In cell cultures and in the absence of cell division, the cccDNA pool appears to be stable [101]. However, cccDNA half-life in vivo can be influenced by different factors, including but not limited to T-cell-derived cytokines. In particular, IFN and lymphotoxin beta receptor (LTβR) were shown to upregulate the expression of proteins from the apolipoprotein B editing complex 3 (APOBEC3) family. APOBEC3s are a family of cytidine deaminase proteins that can act as antiviral proteins by deaminating foreign DNA, converting cytosines to uracils [110], thus causing hypermutations in foreign DNA, including HBV DNA [111]. Lucifora et al. suggested that APOBEC3A/3B upregulation by IFN or LTβR can directly target cccDNA, leading to its non-cytotoxic deamination [112], abasic site formations and degradation.

Previously, CRISPR activation (CRISPRa) was used to achieve an acute and transitory peak of APOBEC expression. In a CRISPRa system, gRNAs targeting the promoter of the gene of interest, in this case APOBEC3, were used to locally recruit a nuclease-inactive Cas9 (dead Cas9) fused to an activation domain [113]. Once located in the promoter of interest, CRISPRa was shown to specifically activate APOBEC3A or 3B expression. Transient APOBEC upregulation led to foreign episomal DNA deamination, omitting foreign integrated DNA. Potentially, CRISPRa could be used to upregulate APOBEC3 expression in infected hepatocytes, leading to an expression peak that could affect HBV viral replication. However, it remains to be determined whether direct cccDNA degradation would be achieved.

Indeed, though DNA deamination via APOBEC3s has been widely studied [114], the downstream processes that lead to deaminated cccDNA destruction rather than repair remain unclear. Using DHBV as a model for HBV cccDNA, Kitamura et al. proposed that the ratio of AID/APOBEC to UNG may be a decisive factor when choosing between deleterious HBV mutations and the generation of new variants [115]. HBV-specific non-cytolytic T-cells were capable of APOBEC3s activation in HBV-infected hepatoma cells or hepatocytes producing viral antigens from integrated HBV DNA, limiting viral infection, strengthening the argument that APOBEC3 proteins affect viral replication [116]. Although these cells would not lead to hepatocyte cytotoxicity, apoptosis followed by cell proliferation was observed in these studies. Therefore, the decrease in cccDNA could also be due to its dilution after cell division. Moreover, recent studies have suggested that cccDNA is not APOBEC’s optimal substrate. Since APOBEC targets preferentially single-stranded DNA (ssDNA), it has been suggested that other replicative intermediates than cccDNA could be more suitable substrates. In vitro studies suggested that HBV DNA deamination occurs during minus-strand DNA synthesis in core particles in the cytoplasm of infected cells instead of via direct cccDNA targeting [94,117]. Furthermore, deamination was preferentially found in HBV ssDNA regions in genomic data from patients [118].

The main drawback of CRISPRa is that overexpression of intracellular deaminases can lead to mutagenesis in the host genome and potentially cancer development [119], thus warranting careful evaluation.

### 5.2. Targeting cccDNA for Epigenetic Silencing (Epigenetic Editors)

HBV protein production was shown to be sensitive to cccDNA methylation state [120], which would make cccDNA an interesting matrix to target with CRISPR interference (CRISPRi) technology. Promoting heterochromatin formation through CRISPRi and achieving epigenetic silencing of cccDNA has not been evaluated so far, but preliminary studies have laid the groundwork for the inhibition of HBV transcription with site-specific modifications.

By fusing the KRAB domain to three zinc-finger proteins, Zhao and colleagues developed the first artificial transcription factor (ATF) to target the Enhancer 1 sequence and the promoter of the HBx gene [89]. A significant reduction in HBx gene transcription was observed in a reporter cell line and in a Hep3B cell line harboring integrated HBV DNA, with cell growth arrest in a Hep3B clone stably expressing the ATF. Using a similar design targeting a different sequence in Enh1, Luo and colleagues extended the characterization of antiviral effects of their ATF in HepG2.2.15, showing reductions in HBV mRNA, HBc and HBx protein, HBeAg and HBV replicative intermediate DNA in vitro, without affecting cell viability [90]. In vivo analyses in HBV transgenic mice confirmed a sustained reduction in HBc protein, HBeAg and HBV replicative intermediate DNA at 28 days post injection. In addition, a consistent effect of the ATF devoid of the KRAB domain was observed, although with a lower efficiency compared to the full ATF, while the non-targeted KRAB effector domain itself did not exhibit a significant effect [90]. The effects of the ATF might therefore be partly attributed to KRAB domain efficiency and also to a potential prolonged steric hindrance in the Enh1 viral sequence, known to be of importance for HBV transcription. Interestingly, the recruitment of another protein, CTCF, in the same sequence was shown to mediate the repression of viral transcription [121]. More potent KRAB domains were developed which could achieve higher repression efficiencies [122].

A zinc-finger-fused to the C-terminus of DNA methyltransferase 3a (XPDnmt3aC) was developed by Xirong and colleagues [84] which targeted the overlapping region between CpG island II and the HBx promoter. De novo methylation of the HBx promoter was achieved, leading to reductions in viral DNA and antigens in a HepG2 cell line. In vivo assessment in an HBV transgenic mouse model showed reductions in HBx and HBsAg protein levels for as long as the expression of XPDnmt3aC was maintained (12 days post injection), followed by a rebound in all parameters, warranting further studies to characterize this transient effect and the associated epigenetic modifications.

In a recent study, Bloom and colleagues [83] engineered an HBV DNA-binding domain of transcription-activator-like effectors (TALEs) fused to a KRAB domain and targeting the S and P ORFs. Interestingly, no specific effect of the single TALEN subunit binding to the DNA was observed, prompting the authors to confirm these findings in an HDI mouse model of HBV infection. Consistent reductions in serum HBsAg, HBV DNA and RNA levels were recorded, along with increased methylation of CpG islands in HBV DNA following hydrodynamic injection. Although no sign of rebound was recorded after 5 days of treatment, characterization of the long-term stability of the methylation pattern will be essential to validate this approach.

Although strategies using epigenetic editors may present an interesting approach for cccDNA targeting, proof-of-concept studies to determine their efficiency in directly targeting cccDNA remain to be conducted. The versatility/cost-effectiveness of CRISPRi may facilitate the testing of a broad range of targets with new Cas proteins with increased PAM flexibility, as well as different effector domains known to repress cccDNA transcriptional activity [123]. In comparison with RNAi strategies, CRISPRi exerts minimal off-target activity [124]; furthermore, precise gRNA localization [125] may maximize repression by steric hindrance, i.e., by competing with viral or host factors for regulatory element accessibility or transcription elongation. As the KRAB domain has been shown to mediate long-range transcriptional repression through heterochromatin spreading [126], additional studies may be of interest to monitor the trans activity of CRISPRi targeting in the host genome and the duration of these effects on cccDNA in the nuclei of infected hepatocytes.

## 6. Remaining Challenges and Perspectives

Future therapies aiming for sterilizing, complete or functional HBV cure are envisaged as combinations of different antiviral approaches. These novel strategies could benefit from approaches that enable direct cccDNA inactivation. Designer nucleases appear to represent a promising strategy that could lead to cccDNA inactivation and/or degradation, thus preventing viral rebound after treatment cessation. The extent to which a reduction in viral parameters should be considered efficient in driving cccDNA elimination is still a matter of debate [4]. Quantitations of the “effect sizes” would differ with respect to several fundamental parameters, including, at least: (i) the number of intrahepatic infected cells and the transcriptional activity of cccDNA; (ii) the HBV integration burden; (iii) the host immune microenvironment; and (iv) the therapeutic strategy employed. Theoretically, combined with highly effective suppression of HBV replication by antiviral agents (more potent NAs, capsid assembly modulators or inhibitors of viral entry), specific disruption of the HBV genome may result in the dilution and, potentially, the eradication of the HBV cccDNA pool. This effect might be enhanced by the use of immunomodulators capable of reinvigorating the HBV-specific host immune response [127].

Nevertheless, several issues still remain to be addressed before considering the clinical application of HBV editing: (i) an exhaustive profiling of possible Cas9 target sites in cccDNA to uncover optimal target sites based on cccDNA accessibility and gRNA binding properties; (ii) delivery of Cas9/gRNA in vivo will require the use of relevant delivery approaches that provide high efficiency and liver specificity: any residual cccDNA could lead to viral rebound and liver repopulation, thus requiring combination strategies to block viral spread from the residual cccDNA; (iii) an extensive study of the fate of the mutated cccDNA variants [95]; and (iv) an extensive genome-wide profiling of off-target effects.

The immunogenicity of gene editing therapies and whether this might be an obstacle to efficient genome editing have also been widely discussed [128]. Pre-existing humoral and adaptive immunity to Cas9 has been reported in a fraction of the population [129]. In mice, pre-existing T-cell response to spCas9 can result in T-cell-mediated clearance of hepatocytes expressing Cas9 via AAV delivery, decreasing the efficacy of the editing [130]. Adaptive immune response has also been observed in mice without pre-exposure to Cas9 [131]. While it did not impact the efficacy of the editing, it remains to be determined whether it could be toxic in HBV patients. The immune response could be diminished by LNP-mediated delivery, as in this case Cas9 is expressed transiently [44].

After the infection of hepatocytes, cccDNA rapidly acquires a dynamic chromatinized structure, which allows different degrees of transcriptional activity in the liver and across the natural history of chronic hepatitis B infection [132,133,134]. Moreover, there is an indication that long-term NAs might lead to cccDNA epigenetic silencing [135]. It has been suggested that SpCas9 and StCas9 have lower on-target activities in genes with restrictive chromatin architectures [136,137] and that high-order chromatin rendered DNA less accessible to nucleases, reducing indel formation [138]. Indeed, a recent work by Wang et al. described the reduced accessibility of transcriptionally inactive cccDNA to dCas9 [139]. Finally, some gRNAs/Cas9 combinations showed higher efficiency in targeting integrated HBV DNA compared to cccDNA, suggesting that cccDNA may be a difficult substrate for targeted mutations and that the targeted regions have to be accurately selected and tested in conditions of varying degrees of cccDNA chromatin accessibility [77].

Another challenge when using HBV-specific genome editing is the mutagenesis rate of HBV, which could increase the chance of escape variants [140]. Additionally, HBV genomic heterogenicity, illustrated by its different genotypes or sub-genotypes, may represent an obstacle to applying gRNAs in patients in whom HBV exists as a quasispecies. Both these issues could be overcome by the design of gRNAs targeting conserved regions in the HBV genome: this will not only be of benefit to the patients infected with different viral genotypes/variants but could also help to avoid the emergence of escape mutants. A complementary strategy could be to use a combination of the gene editing reagents (such as several gRNAs) targeting all pre-existing HBV variants. Similarly, a combination of shRNAs prevented the selection of resistant HBV variants in a human liver chimeric mouse model [141]. An alternative strategy could be to combine gene editing with other antiviral agents targeting the HBV life cycle at different steps.

Altogether, compared with recent reviews of the topic [142,143], significant advances have been made in the last few years regarding the development of cccDNA targeting strategies and the comprehension of cccDNA fate after editing. Major concerns regarding host genome instability issues that might follow DSBs in HBV-integrated sequences prompted the use of newly developed base editing strategies which have shown promising preliminary results. Moreover, recent significant advances in liver-targeted delivery strategies in clinical settings for the cure of severe genetic diseases ([46] and NCT05398029) have tremendously expanded the potential use of DNA editing strategies. As the current HBV standard of care, NUCs have relatively safe profiles, emerging gene editing therapies would likely have to closely match that bar. Additional guidelines could come from interferon treatment, which is associated with more adverse effects and is still recommended in certain cases, with a functional cure rate of about 1–3% [2]. In order to progress, gene editing would need to be made much more efficient in providing functional cure and be associated with fewer side effects compared with interferon treatment. Additional safety requirements would need to be tailored for particular gene therapies, including comprehensive assessments of off-target profiles. Only a thorough assessment of the safety of these therapies will determine whether they will be able to match the requirements for use in combination therapies to achieve HBV cure.

## Figures and Tables

**Figure 1 viruses-14-02654-f001:**
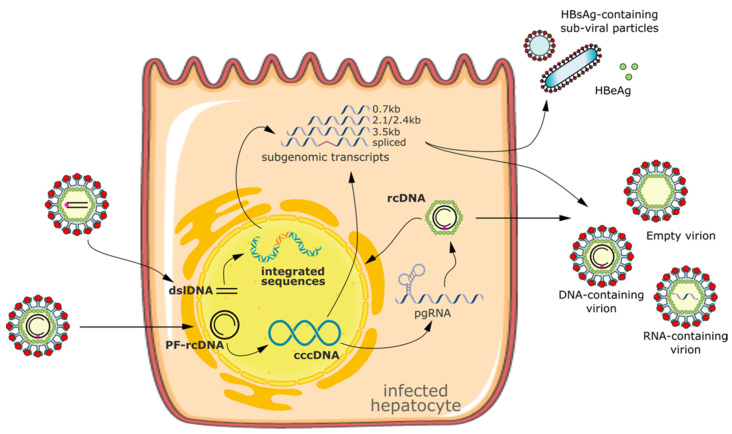
Schematic representation of HBV lifecycle and HBV genomic forms potentially targeted by gene editing. Once it has entered into the hepatocytes, the rcDNA and the dslDNA are released into the nucleoplasm, where PF-rcDNA is converted into cccDNA, while dslDNA might represent the substrate for HBV integration into the host genome. cccDNA is the only template for viral pregenomic RNA (pgRNA) and all the viral subgenomic mRNAs, while HBsAg and a truncated but potentially functional HBx form can be produced from integrated HBV DNA. pgRNA is retrotranscribed by the viral polymerase into rcDNA. A fraction of rcDNA is packaged and released, forming new infectious particles (i.e., Dane particles), while some will be transported to the nuclei to replenish the cccDNA pool. rcDNA, dslDNA, cccDNA and HBV integrated sequences have been demonstrated to be targeted by gene editing. It remains to be elucidated whether the rcDNA contained in the neo-formed cytoplasmic nucleocapsids could also be modified. rcDNA, relaxed circular DNA; PF-rcDNA, protein-free relaxed circular DNA; dslDNA, double-strand linear DNA; cccDNA, covalently closed circular DNA.

**Figure 2 viruses-14-02654-f002:**
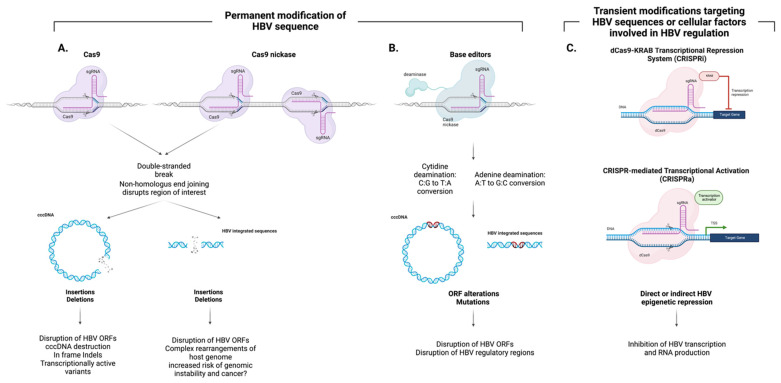
Cas9-based approaches for HBV genome editing. (**A**) Standard Cas9 and Cas9-nickase act through generating a double-stranded break (DSB), which may lead to large deletions and genomic rearrangements, especially in cases where several locations in the genome are targeted. In addition to an episomal cccDNA, chronic hepatitis B patients typically have multiple copies of HBV DNA integrated into their genome, so the possibility of chromosomal translocations is significantly increased. (**B**) Newer editing systems, such as base editing, result in permanent modification of the genome without generating DSBs. (**C**) Alternative approaches rely on the fusion of catalytically inactive Cas9 (dCas9) to transcriptional activators or repressors. These strategies aim at transiently affecting HBV RNA production and thus replication by modifying the HBV epigenome either directly or indirectly by targeting HBV regulators. Image created with BioRender.com.

**Table 1 viruses-14-02654-t001:** General characteristics of programmable nucleases.

	Meganucleases	ZNFs	TALENs	CRISPR/Cas9
DNA binding interface	Protein-DNA	Protein-DNA	Protein-DNA	RNA-DNA
Composition	DNA binding and cleavage domain	Zinc-finger domain + FokI nuclease	TALE-DNA-binding domain + FokI nuclease	crRNA, Cas9 protein
Recognition sequence	12–45 bp	18–36 bp	30–40 bp	22 bp
Restrictions on target site		G-rich	Start with T and end with A	PAM at the end of target sequence
Cost ^1^	High	High	Middle	Low
Off-target events	Low	Comparable	Comparable	Comparable
Delivery ^2^	Easy	Easy	Difficult	Moderate
Multiple targeting	Difficult	Difficult	Difficult	Easy
Sensitivity for methylation of DNA target	High	High	High	Low
Cytotoxicity	Variable to high	Variable to high	Low	Low

^1^ Based on the manufacturing of the editor. ^2^ Based on the size of the editor.

**Table 2 viruses-14-02654-t002:** Approaches for in vivo delivery of gene editors.

Viral Delivery	Pros	Cons
AAVs	Minimal risk of integration into the host genome Mild host immune responses Can be pseudotyped	Limited size of the cargo that can be packaged Continuous expression of editors, increasing risk of off-target effects Limited possibility to redose, due to the immune response
AdVs	Minimal risk of integration Not limited by the size of cargo	Undesirable immune responses Undesirable side effects Not suited for targeted delivery Continuous expression of editors, increasing risk of off-target effects
Lentiviruses	Can be pseudotyped Not limited by the size of cargo	Integrated into the host genome Continuous expression of editors, increasing risk of off-target effects
Non-viral delivery		
LNPs	Minimal safety and immunogenicity concerns Possibility of re-dosing Can be targeted to specific cell populations Transient expression of editors, thus minimizing off-target effects	Low transfection efficiency

AdVs, adenoviruses; AAVs, adeno-associated viruses; LNPs, lipid-based nanoformulations.

**Table 3 viruses-14-02654-t003:** Summary of published HBV editing strategies.

Model	Editing Strategy	Delivery Strategy	References
**In Vitro Models**			
Transfection of cell lines with HBV-expressing plasmids	TALENs	Plasmid transfection	Bloom, Mol Ther 2013 [71] Chen, Mol Ther 2014 [72] Dreyer, BBRC 2016
	ZFNs	Plasmid transfection	Cradick, Mol Ther 2010 [73]
	CRISPR/Cas9	Plasmid transfection	Lin, Mol Ther Nucleic Acids 2014 [74] Dong, Antivir Res 2015 [75] Liu, J Gen Virol 2015 [76] Ramanan, Sci Rep 2015 [77] Wang, World J Gastroenterol 2015 [78] Zhen, Gene Ther 2015 [79] Zhu, Virus Res 2016 Liu, Antivir Res 2018 [80] Yan, Frontiers Microbiol 2021
	CRISPR/Cas9 nickase	LLNs	Jiang, Cell Res 2017
		Plasmid transfection	Karimova, Sci Rep 2015 [81] Sakuma, Gene Cells 2016 Kurihara, Sci Rep 2017 [82]
	TALEN-KRAB	Plasmid transfection	Bloom, BMC Infect Dis 2019 [83]
	ZFN-Dnmt3a	Plasmid transfection	Xirong, Biochemistry 2014 [84]
Cell lines harboring the integrated HBV genome	TALENs	Plasmid transfection	Bloom, Mol Ther 2013 [71] Dreyer, BBRC 2016
	ZFNs	AAV transduction	Weber, Plos One 2014
	CRISPR/Cas9	Plasmid transfection	Dong, Antivir Res 2015 [75] Wang, World J Gastroenterol 2015 [78] Zhen, Gene Ther 2015 [79] Liu, Antivir Res 2018 [80] Li, Front Cell Infect Microbiol 2017 [85] and Int J Biol Sci 2016 [86]
		Lentiviral transduction	Ramanan, Sci Rep 2015 [77] Kennedy, Virology 2015 [87] Kayesh, Virus Res 2020 [37]
		AAV transduction	Scott, Sci Rep 2017 [88] Kayesh, Virus Res 2020 [37]
	CRISPR/Cas9 nickase	Lentiviral transduction	Karimova, Sci Rep 2015 [81]
		Plasmid transfection	Kurihara, Sci Rep 2017 [82]
		AdV transduction	Schiwon, Mol Ther Nucleic Acids 2018 [40]
	ZFN-KRAB	Lentiviral transduction	Zhao, J Biomol Screen 2013 [89]
		Plasmid Transfection	Luo, Int J Mo Med 2018 [90]
	CBEs	Lentiviral transduction	Yang, Mol Ther Nucleic Acids 2020 [91]
HBV infection system	Meganuclease	LNPs	Gorsuch, Mol Ther 2022 [92]
	CRISPR/Cas9 nickase	Lentiviral transduction	Karimova, Sci Rep 2015 [81] Kurihara, Sci Rep 2017 [82]
	CRISPR/Cas9	Lentiviral transduction	Ramanan, Sci Rep 2015 [77] Seeger, Mol Ther Nucleic Acids 2014 [93] and 2016 [94] Kennedy, Virology 2015 [87]
		AAV transduction	Scott, Sci Rep 2017 [88] Kayesh, Virus Res 2020 [37]
		AdV transduction	Schiwon, Mol Ther Nucleic Acids 2018 [40]
		RNP transfection	Martinez, mBio 2022 [95]
	CBEs	Lentiviral transduction	Zhou, Hepatol Commun 2022 [96]
**In Vivo Models**			
HDI with HBV-expressing plasmids or precccDNA	TALENs	Plasmid HDD	Bloom, Mol Ther 2013 [71] Chen, Mol Ther 2014 [72]
	CRISPR/Cas9	Plasmid HDD	Lin, Mol Ther Nucleic Acids 2014 [74] Dong, Antivir Res 2015 [75] Liu, J Gen Virol 2015 [76] Ramanan, Sci Rep 2015 [77] Zhen, Gene Ther 2015 [79] Kurihara, Sci Rep 2017 [82] Li, Int J Biol Sci 2016 [86]
		LLNs IV injection	Jiang, Cell Res 2017
		AAV HDD	Liu, Antivir Res 2018 [80] Yan, Frontiers Microbiol 2021
		Plasmid HDD	Bloom, BMC Infect Dis 2019 [83]
AAV-HBV mice	Meganuclease	LNPs	Gorsuch, Mol Ther 2022 [92]
HBV transgenic mice	CRISPR/Cas9	Plasmid HDD	Zhen, Gene Ther 2015 [79] Zhu, Virus Res 2016 Li, Int J Biol Sci 2016 [86]
		AAV IV injection	Li, Front Immunol 2018 [97]
	ZFNs-KRAB	Plasmid HDD	Luo, Int J Mo Med 2018 [90]
	ZFNs-Dnmt3a	Plasmid HDD	Xirong, Biochemistry 2014 [84]
Human hepatocyte chimeric mice with HBV infection	CRISPR/Cas9	AAV IV injection	Stone, Mol Ther Methods Clin Dev, 2021 [98] Kayesh, Virus Res 2020 [37]
Non-human primate AAV-HBV model	Meganuclease	LNPs	Gorsuch, Mol Ther 2022 [92]

HDD, hydrodynamic injection; LNPs, lipid nanoparticles; LLNs, lipid-like nanoparticles; IV, intravenous; RNPs, ribonucleoparticles; AdV, adenovirus; TALENs, transcription-activator-like effector nucleases; ZFNs, zinc-finger nucleases; KRAB, Krüppel-associated box; Dnmt3a, DNA Methyltransferase 3 Alpha.

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
