# Peer review of "Gene Editing Technologies to Target HBV cccDNA"

_viruses, 2022, doi:10.3390/v14122654_

Round 1

Reviewer 1 Report

In this review by Martinez et al., the authors outline the use of gene editing technologies to target HBV cccDNA, the ultimate goal of thereapeutics in the hepatitis B field. The review is thorough and appears to cite the majority of studies in the field. Some sections are unclear with regard to specific point and the narrative is not entirely clear at all points. The following issues should be addressed before the paper be considered for publication:

11.       There have been previous reviews on this topic by not only other groups (e.g. doi: 10.3390/genes9040207; doi:  10.4254/wjh.v7.i2.150; doi: 10.20944/preprints201710.0065.v1; and https://doi.org/10.1016/j.virusres.2017.06.010; https://doi.org/10.1016/j.virusres.2017.01.003; DOI: 10.3390/ijms160817589; and doi:10.1038/gt.2014.94; doi: 10.3390/ijms160817589; PMID: 26463095 and doi: 10.1007/978-1-4939-2432-5_2). It is unclear to me how this review differs or significantly improves on these previous publications.

22.       Table 1 is confusing:

a.       What does DNA binding mean?

b.       “Targeted sequence” should be “recognition sequence” as the nucleases do not cut the entire sequence

c.       Change “target site” to “restrictions on target site”

d.       Unclear what the “cost” and “delivery” is based on

e.       Unclear to what extent methylation attenuates the

33.       There are long descriptions of many delivery technologies described in this review (e.g. AAV, LNPs, etc.) but it is difficult to determine from this what advantages (and disadvantages) does one have over the other? It would be clearer with a table. How do these impact on their role to therapeutically treat HBV or eliminate cccDNA?

44.       It is unclear what section 3 adds to the overall review targeting cccDNA.

55.       With such a glut of primary studies, the authors do not make clear why the use of these strategies as therapeutics has not appeared to make significant headway. It would be good to clarify somewhere what factors (e.g. delivery, toxicity, cancer risk, effect size, viral targets, resistance and rebound) have been achieved to an appropriate level for therapies and which still have not – in which case provide the estimated distance still to cover.

66.       Effect size is a probably a significant limitation. The authors in many parts of the review do not explicitly mention the effect size of the reduction in HBV markers (such as cccDNA number or antigen expression). Further, the authors give no context on what kind of effect size is necessary to eliminate cccDNA.

77.       Section 4.1 is difficult to follow as it simultaneously explains models for HBV, technologies used for gene editing, results from various gene editing studies, and mechanisms of the editing.

88.       Figure 1 text is too small to read.

99.       It might also be worth mentioning targeting of other cellular factors with CRISPR – (e.g. DOI: 10.1016/j.omtm.2021.11.002)

110.   It is unclear why CRISPRa is better than simply expressing a transgene.

111.   Isn’t the main drawback for the CRISPRa study mentioned the effect seems to be on rcDNA not cccDNA (in addition to the mutagenesis)? What is the antiviral effect size of this approach?

112.   Section 6, appears to be poorly structured and simply a list of drawbacks.

a.       How does gene editing synergise with other drugs to achieve good treatment response/cure? It might be worth explicitly mentioning how it helps elimination in a unique way.

b.       Are multiple treatment of CRISPR therapy achievable? Would it help?

c.       Will it be possible to exhaustively profile possible Cas9 sites given everyone has a unique genetic makeup? What level of off-target effect is acceptable?

d.       It is unclear from this section what the authors define as safety of these therapies. It might be appropriate to include affected community consultation to define this balance between possible risks and benefits.

Author Response

Please, see attached document

Reviewer 2 Report

This is a timely and comprehensive review that summarizes the up-to-date information about the development of DNA editing technologies to target HBV cccDNA, the persistent form of HBV genome and the major obstacle to a cure of chronic hepatitis B. Although it remains at the preclinical stage, cccDNA-targeted gene editing approaches represents a feasible option to directly destruct and/or inactivate cccDNA, so far there is no other mean that can, either chemically or immunologically, achieve such clear and specific inhibitory effects on cccDNA in experimental systems. In addition, the cccDNA editing approach keeps being improved following the further development of gene editing towards better specificity, efficiency, and safety in general. In these regards, this review article is very informative to a broad audience interested in treatment of infection by DNA viruses and retroviruses. Another major merit of this article is the assessment of pros and cons of currently available cccDNA-targeted editing technologies, the actual HBV DNA species (cccDNA, core DNA intermediates, integrated DNA) being targeted in previous studies were also discussed. The manuscript is also well written. I only have a couple of minor suggestions for the authors.

1. It would be beneficial to the readers if the authors could include an illustration of all the HBV DNA species that may be targeted by the designer nucleases in virally infected cells.

2. The reported immunogenicity of Cas9 protein and the preexisting immunity against Cas9 therapy can be discussed. 

Author Response

please, see attached document

Round 2

Reviewer 1 Report

See attached file (comments highlighted yellow)

Author Response

Please, see attached document
